

# Effect of birdsongs and traffic noise on pedestrian walking speed during different seasons

Marek Franěk, Lukáš Režný, Denis Šefara and Jiří Cabal

Faculty of Informatics and Management, University of Hradec Králové, Hradec Králové, Czech Republic

## ABSTRACT

Many studies have explored the effects of auditory and visual stimuli on the perception of an environment. However, there is a lack of investigations examining direct behavioral responses to noise in specific environments. In this study, a behavioral variable, walking speed, was analyzed, as a response to the sounds and visual features of a specific environment. The study examined the effects of birdsongs compared to traffic noise on walking speed in a real outdoor urban environment. It was supposed that the interaction of audition and vision in the perception of an environment may also be shaped by the perceived congruence of the visual and auditory features of the environment. The participants ($N = 87$ and $N = 65$), young university students, walked along a 1.8-km urban route. They listened to a soundtrack of crowded city noise or birdsongs, or they walked in the real outdoor environment without listening to any acoustic stimuli. To investigate the effect of the congruence between acoustic and visual stimuli, the experiment was conducted in two different seasons (fall and spring). The results did not show significant differences between the crowded city noise condition and the real outdoor condition. Listening to the soundtrack with birdsongs decreased walking speed, but this effect was significant only in the experiment conducted in spring. These findings can be explained in terms of the congruence between the sounds and the visual environment. The findings raise questions regarding the restorative function of urban greenery during different seasons.

## INTRODUCTION

Several decades ago, some cultural anthropologists created the concept of pace of life. This concept proposes that daily life follows a certain rhythm. The pace of life has been studied in large cities across geographic locations, operationalized as the temporal aspect of various daily behaviors and activities. Early research examined walking speed, the speed with which postal clerks completed a simple request, time punctuality, etc., in downtown areas of large cities (*Werner, Altman & Oxley, 1985*). More recent studies searched, for instance, for a rate of tweets on the social media platform Twitter as a function of population density (*Gross, Murthy & Varshney, 2017*). A fast pace of life may be a response to stimulatory overload and various urban stressors, including crowding and traffic noise (e.g., *Bornstein*

Corresponding author
Marek Franěk, marek.franek@uhk.cz

& Bornstein, 1976). There is evidence that stressful sounds increase arousal, which activates the internal clock. They may also result in the acceleration of walking speed (*Ozel, Larue & Dosseville, 2004*).

Urban pedestrian walking speed was investigated mainly in the 1970s and the 1980s (*Bornstein & Bornstein, 1976*; *Kirkcaldy, Furnham & Levine, 2001*; *Levine & Bartlett, 1984*; *Lowin et al., 1971*; *Walmsley & Lewis, 1989*). Importantly, the results of more recent research suggests that the walking speed in large cities may gradually increase (*Wiseman, 2007*). Previous investigations also documented the negative health consequences of a fast pace of life, including walking speed. It was shown that a fast pace of life is associated with a high likelihood of heart attacks (e.g., *Evans, 1984*; *Levine & Bartlett, 1984*). A fast movement speed and speed of other daily activities has been interpreted in parallel to Type A behavior patterns (a potential risk factor for heart disease) (*Levine et al., 1989*). It seems that a fast walking speed as part of the overall pace of life is a potential health risk factor. Thus, investigations of these phenomena have some relevance. Due to the sedentary lifestyles of many city residents, it is recommended that they participate in, moderate-intensity physical activities every day. Walking is an ideal example of moderate-intensity physical activity, and it has positive consequences for physical health (e.g., *Morris & Hardman, 1997*; *Hanson & Jones, 2015*). A walk in an urban green environment also has many psychological benefits (e.g., *Berman, Jonides & Kaplan, 2008*; *Bratman, Hamilton & Daily, 2012*; *Hartig et al., 2003*). Of course, it cannot be claimed that walking fast generally has negative health. A fast walking pace is not an undesirable behavior if it is a part of some sport or recreational activity, for instance. However, the above-described findings addressed the everyday walking speed of urban pedestrians in the context of the overall pace of life as a response to stimulatory overload and various urban stressors. In the present study, we examine to what extent walking speed during an urban walk can be affected by noise or, on the other hand, listening to natural sounds.

## Walking pace and environmental sounds

Current investigations provided some evidence that noise exposure might influence the walking pace. A recent study by *Maculewicz, Erkut & Serafin (2016)* clearly showed that the sound characteristics of specific environments affect walking pace in those environments. The participants were asked to listen to the sounds of a seashore, a busy street, a restaurant, and busy offices and simultaneously walk on an aerobic stepper. The results revealed that the seashore and restaurant sounds elicited a slower pace than the sounds of streets and offices. Interestingly, the results documented not only the effect of traffic noise on the increase in walking pace but also indicated that listening to nature sounds may result in a decrease in walking pace. In our previous study (*Franěk, 2013*) participants were asked to walk along a route located in an urban area with different environmental features. The participants tended to walk statistically significantly faster in areas without greenery and with more traffic and higher perceived noise than in sections with greenery and less traffic and perceived noise. Our previous study (*Franěk et al., 2018*) found that listening to a soundtrack with traffic noise while walking on an urban route increased the average walking speed. Conversely, listening to a soundtrack with natural sounds (birdsongs)

decreased the walking speed in contrast to the control condition, but this difference was not significant. Moreover, under both conditions, the walking speed was lower in areas with a greater amount of perceived naturalness than in areas with less greenery.

## Auditory and visual interactions

There is a large body of studies on auditory and visual interactions. One line of research, based on the subjective evaluation of a combination of photographs of diverse environments and sounds, showed that certain sounds can influence the evaluation of physical environments. In a pioneering study, *Anderson et al. (1983)* showed that natural and animal sounds had enhancing effects on evaluations of wooded natural and residential sites, while other sounds had detracting effects on evaluation of the same sites. More recently, *Gan et al. (2014)* found that anthropogenic sounds (vehicle alarms, motorcycle rumbling, the roar of engineering machinery, adult voices) had a negative impact on landscape preference, as opposed to biological and geophysical sounds. The authors also showed that acoustic preference played a much more important role in landscape evaluation than visual preference. Furthermore, a set of studies has examined the effect of anthropogenic sounds on the ratings of landscapes in natural parks. *Mace, Bell & Loomis (1999)* examined the effect of helicopter tour noise on evaluations of scenic overviews in a national park. The negative experiential effect was more strongly associated with soundscapes that included helicopter noise than with purely natural soundscapes. *Benfield et al. (2010)* reported that anthropogenic soundscapes (helicopter noise, airplane overflights, motorized ground vehicles, and human voices) were each responsible for detriments to the visual assessment of the landscapes shown. *Weinzimmer et al. (2014)* replicated previous studies by showing that motorized recreation noise had detrimental effects on the ratings of both aesthetic and affective dimensions.

Conversely, another line of research showed that the evaluation of sound environments can be affected by co-occurring visual settings. *Viollon, Lavandier & Drake (2002)* found that the more urban the visual setting, the more negative the sound ratings of the environment. *Lee, Hong & Jeon (2014)* explored how participants assessed rural soundscapes featuring high-speed train noise. The noise from the high-speed train was rated as less annoying if the sound was presented with a picture containing a higher percentage of natural features. The same effect documented also surveys of residents exposed to noise in their neighborhood. *Li, Chau & Tang (2010)* studied the effects of annoyance modifiers on residents of high-rise buildings overlooking urban parks and wetlands. Their results indicated that the perception of greenery considerably reduced noise annoyance and concentration problems Having a window facing a yard and a view of water or green space is associated with substantially reduced noise annoyance and concentration problems (*Bodin et al., 2015*). Consistently, *Van Renterghem & Botteldooren (2016)* reported that the extent to which vegetation is visible through the living room window was a strong predictor of self-reported noise annoyance among residents with high exposure to traffic noise. *Leung et al. (2017)* found that views of the sea, urban river, or greenery could reduce the probability of invoking a high noise annoyance response from residents living in high-rise buildings, while views of a noise barrier could increase the

probability. Views of greenery had a stronger noise moderation capability than views of the sea or urban river. *Cassina et al. (2017)* investigated tranquility ratings in urban areas judged on the basis of visual and auditory elements. The authors showed that the positive sound sources increased statistically significantly the perceived tranquility, while negative sound sources decreased it. However, visual elements had only negative effects on the tranquility score.

*Van Renterghem (2019)* summarized potential explanations for these effects. First, it was shown that nature sounds are preferred over anthropogenic ones, namely over vehicle and construction sounds (*Hong & Jeon, 2013*; *Krzywicka & Byrka, 2017*; *Medvedev, Shepherd & Hautus, 2015*; *Ratcliffe, Gatersleben & Sowden, 2016*; *Yang & Kang, 2005*). Several studies documented that after inducing psychological stress, physiological recovery of sympathetic activation is faster during exposure to pleasant nature sounds than to sound perceived as less pleasant (*Alvarsson, Wiens & Nilsson, 2010*; *Annerstedt et al., 2013*; *Medvedev, Shepherd & Hautus, 2015*). Natural sounds are considered the most complex and informational sound types. They can provide a large amount of information pertaining to species, season, and temporality (*Pijanowski et al., 2011*), and signify an actual living or vital environment (*Ratcliffe, Gatersleben & Sowden, 2013*). Importantly, natural soundscapes can provide restorative benefits independent of those produced by visual stimuli (*Benfield et al., 2014*). Bird sounds are ranked at the top of the desired natural sounds in an urban environment; they outrank the other sounds in terms of their pleasantness (for review, *Van Renterghem, 2019*). Bird sounds are most commonly associated with perceived stress recovery and attention restoration (*Ratcliffe, Gatersleben & Sowden, 2013*), and they are associated with green spaces, spring, and summer (*Ratcliffe, Gatersleben & Sowden, 2016*). Second, some studies have documented that the invisibility of a source of noise among vegetation decreases perceived noisiness (e.g., *Bangjun, Lili & Guoqing, 2003*; *Sun et al., 2018*; *Watts, Chinn & Godfrey, 1999*). Finally, vegetation is often a source of natural sounds. Moreover, natural sounds increase one's feeling of the presence of nearby nature, even when it is not directly visible.

## Restorative environment

Environmental psychology presents a large body of evidence that direct and indirect exposure to nature positively affects humans. There is considerable evidence that living in green areas or even having a view of nature results in decreased stress. It was observed that the residents of neighborhoods with a high amount of greenery have relatively low chronic stress (e.g., *Hartig et al., 2011*; *Nilsson & Berglund, 2006*; *Ward Thompson et al., 2012*). Stress recovery was observed to be more rapid in people who viewed natural scenes than in people who viewed urban scenes (e.g., *Ulrich et al., 1991*). Furthermore, a large body of studies has shown that viewing a surrogate image of nature (photographs, videos, slides, window views, and virtual nature scenes) resulted in decreased stress and increased positive emotions (e.g., *Brown, Barton & Gladwell, 2013*; *De Kort et al., 2006*; *Felnhofer et al., 2015*; *Hartig et al., 1999*; *Jiang, Chang & Sullivan, 2014*; *Ulrich, Simons & Miles, 2003*; *Valtchanov, Barton & Ellard, 2010*).

As previously mentioned (e.g., *Anderson et al., 1983*; *Gan et al., 2014*), natural and animal sounds had enhancing effects on the evaluations of the restorative potential of wooded natural and residential sites. It is known that people use various commercially produced recordings of natural sounds for their relaxation. Recently, virtual reality relaxation applications have also been available (e.g., *Lindner et al., 2019*). There is research evidence of the effects of listening to natural sounds on relaxation. Such studies documented enhanced stress recovery during exposure to pleasant nature sounds (e.g., *Alvarsson, Wiens & Nilsson, 2010*; *Annerstedt et al., 2013*; *Medvedev, Shepherd & Hautus, 2015*). Although diverse natural sounds may be used (e.g., ocean, birds, rain, nighttime in the jungle, water, and waterfall sounds), bird sounds have specific effects. Several studies have documented that this type of natural sound is most commonly associated with green spaces, spring, and summer (*Ratcliffe, Gatersleben & Sowden, 2016*) and with attention restoration and perceived stress recovery (*Ratcliffe, Gatersleben & Sowden, 2013*). Urban settings with birdsong are more highly evaluated than urban settings alone, and birdsong contributes to the positive ratings associated with urban green space (*Hedblom et al., 2014*).

## Congruence of the visual and auditory features of the environment

The interaction of audition and vision in the perception of an environment may also be shaped by the perceived congruence of the visual and auditory features of the environment. "Congruence" between the two types of stimuli means that their combination makes sense because people expect specific sounds to occur in each environment that is congruent with the physical features of the environment (*Bruce & Davies, 2014*). In the study by *Carles, Bernáldez & Lucio (1992)* visual stimuli (village, stream, park with children, steppe, empty park, and residential neighborhood) and corresponding sound stimuli were presented in varying combinations. The authors found that the congruence between sound and image influenced the preferences. *Brambilla & Maffei (2006)* showed that in parks, the more that sounds were congruent with expectations, the less they evoked annoyance. *Hedblom et al. (2017)* reported that people gave higher evaluations to natural sounds in areas that they considered highly natural, such as urban woodlands, compared with less natural areas, such as parks, allotments. and lawns. A study by *Zhao, Xu & Ye (2018)* demonstrated that appropriate combinations of congruent acoustic and visual stimuli increased the perceived restorative potential of an environment. For instance, a landscape containing natural water and high plant coverage matches the visual association with a bird singing, and combining wind sounds with a landscape with high vegetation coverage increases the restorative quality of that environment. Furthermore, some studies suggest that evaluations of natural environments are more sensitive to visual-auditory incongruence than evaluations of urban environments. *Ge & Hokao (2005)* found that the sounds of transportation were more disliked in a natural landscape than in ordinary urban street environments where they were more congruent. Consistently, *Jahncke, Naula & Eriksson (2015)* examined the combined effect of various acoustic stimuli (nature sounds, quiet broadband noise, office noise) and visual settings (office and urban nature environments) on perceived restoration. They found that a picture of nature was more sensitive to the influence of auditory stimuli than a picture of an office.

## The current study

The objective of the present study was to analyze the combined effect of diverse sounds and visual environmental features on walking speed in certain outdoor environments. The present study aimed to enhance these findings by conducting two experiments in different environments and seasons.

In a previous study (*Franěk et al., 2018*), participants were asked to walk along an urban route while listening to soundtracks with intense crowded city traffic noise (noise from motorized vehicles, engine sounds, automobile horns, and human voices) or birdsongs from headphones, or they walked without listening to any acoustic stimuli in the control condition. The participants listening to traffic noise walked significantly faster on the route (mean walking speed was 1.65 m/s) than those listening to forest birdsong sounds (mean walking speed was 1.53 m/s). The participants listening to the birdsong walked slightly slower than those under the control condition (mean walking speed was 1.58 m/s), but this difference was not statistically significant. Simultaneously, the walking speed in all conditions was influenced by the environmental features of the route. In areas with more natural features, the participants walked more slowly than in areas with less greenery. Consistent with previous findings, the participants liked the environment more in the absence of noise or in the presence of birdsongs. Importantly, the previous experiment not only documented the negative effect of noise but also showed that relaxation sounds may decrease walking speed. This study also raised questions regarding the congruence between sounds and the visual environment. Do nature sounds (forest birdsong) have an effect on the deceleration of walking speed only in congruent environments? Specifically, the question we sought to answer was whether listening to birdsongs that are commonly associated with spring and summer had similar effects in May and November. In the previous study (*Franěk et al., 2018*), the walking route was located in green areas with a relatively small amount of traffic. The first part of the route was a street with driving cars located along a small park or a meadow with threes; the second part was a dense oak alley that led out into the street with traffic. What is the effect of spring birdsongs when walking along a busy road or when walking in the fall?

The current study continues the previous investigation (*Franěk et al., 2018*) by using an identical methodology but in different locations and in two diverse seasons. While the previous experiment was conducted in a relatively calm area with a greater amount of greenery, in the current study, the participants were asked to walk along a route that contained areas near both a busy road and a calm alley to test, whether the observed associations between sounds and walking behavior would also occur in a different environment. Specifically, our objective was to determine whether the effect of nature sounds (birdsongs) also occurs in an environment with a high level of traffic, which may not be perceptually congruent with nature sounds. Furthermore, to study the effects of the environmental features and perceived congruence between the sounds and the environment more precisely, the experiments were conducted on an identical route during two vegetation periods - in November and in May. As in the previous study, we used three experimental conditions: (1) listening to a soundtrack with birdsongs, (2) listening to a soundtrack with crowded city noise, and (3) listening to no soundtrack (control

condition). However, walking without listening to a soundtrack cannot represent a true control condition in the current study because, in contrast with the outdoor environment used in the previous study, high levels of noise occur in some areas. On the other hand, it is almost impossible to have a perfect control condition in a real outdoor environment. Thus, the control condition in the present experiment represents the real outdoor noise. In summary, the aim of the current study was to investigate the effects of listening to diverse environmental sounds on walking speed while walking on a route in a real outdoor urban environment in two different seasons. Consistent with previous findings, it was presumed that listening to the soundtrack with intense crowded city noise would increase walking speed, while listening to the soundtrack with nature sounds would decrease walking speed. We also supposed that listening to the soundtrack with intense crowded city noise may increase walking speed in comparison to that when listening to real outdoor noise. Furthermore, the effect of congruence between the environment and sound was examined. It was hypothesized that the effect of bird sounds would be more salient in the spring than in the fall because of congruence between the sounds and the environment. Finally, the effects of both soundtracks on the evaluation of the walking route and the walking experience were examined. It was hypothesized that the walk would be evaluated as more pleasant while listening to the soundtrack with birdsongs than when listening to the soundtrack with traffic noise and under the real outdoor noise condition.

## MATERIALS & METHODS

### Stimulus material

There were three conditions. The participants listened to a soundtrack with relaxation nature sounds or a track with traffic noise; they did not listen to any soundtrack under the real outdoor noise condition. The soundtrack from the video "Forest Birdsong - Relaxing Nature Sounds - Birds Chirping", available on YouTube (https://www.youtube.com/watch?v=Qm846KdZN_c), was selected as the nature sound. The track consists of the sound of birds singing. The track lasted approximately 2 h. The soundtrack from the video "Hectic Kolkata (Calcutta)—India", available on YouTube (https://www.youtube.com/watch?v=IFc2KhKLiho), was selected as the crowded city noise. The track contains noise from motorized vehicles, engine sounds, automobile horns, and human voices. The track was modified using the software Audacity because of its short length to have a length of 38 min and 43 s, corresponding to the length of the participants' walk. The sound levels of the tracks were adjusted to a comfortable level and did not differ among participants. The mean sound pressure levels were as follows: forest birdsong—56 dB(A) (a relatively high magnitude of the mean sound pressure level reflects higher differences between thesound levels of a silent background and the birdsongs), and crowded city noise—53 dB(A). The participants listened to the tracks, which were played on a cell phone.

The sounds were listened to through lightweight headphones Genius HS-M200C. For safety reasons, headphones did not entirely mask sounds from the outside. Participants assigned to the real outdoor noise condition did not wear headphones while walking.
## Walking route

The experiment was conducted in Hradec Králové in the Czech Republic. This city is located in the northeastern part of the Czech Republic and has approximately 100,000 inhabitants. The walking route was a circuit with a length of approximately 1.8 km (see Fig. 1). The route was located in areas with minimal movement of other pedestrians; thus, walking speed was not affected by the flow of other pedestrians. Eight sections were chosen for the analysis of walking speed. The sections were chosen to provide a direct route and avoid crossroads or other obstacles. The lengths of the sections ranged from 45 to 100 m. The first part of the route was an oak alley along a river (sections 1–3). There was no street with traffic close to this area. Traffic noise in these sections was approximately Lday = 55–60 dB(A) (*Hlukové mapy, 2012*). Section 4 was located still in the oak alley, but it was closer to a road with dense traffic. Traffic noise was approximately Lday = 60–65 dB(A) (*Hlukové mapy, 2012*). Sections 5 and 6 went along a busy road, where cars, buses, and trucks were driving. On one side of the route, there was a busy road, while on the other side, there were low trees and open areas with a lawn. Traffic noise in these sections was approximately Lday = 75–80 dB(A) (*Hlukové mapy, 2012*). The final part of the route (sections 7 and 8) went along a different street with less traffic. On one side of the route was a street, while on the other side, there were low trees and open areas with a lawn. The traffic noise in section 7 was approximately Lday = 65–70 dB(A), and in section 8, it was approximately Lday = 55–60 dB(A) (*Hlukové mapy, 2012*).

## Measurement of walking speed

The participants walked with a small video camera (i.e., a Sony Bloggie MHS-PM5K) on a belt around their waist (size $19 \times 108 \times 55$ mm, weight 110 g). The environment, the participant's feet, and the participant's arms were captured through a fish eye lens. Beginning and end of each section of the route was indicated by a line drawn on the sidewalk. An evaluator marked two frames of the video recording to create the beginning and end of the annotation for each particular track section in the software Elan (see https://tla.mpi.nl/tools/tla-tools/elan/). Every annotation represented the entire section of the track, so that the extent of time subjects spent there could be determined. This enabled to calculate the average speed reached by the participants in all sections.

## Evaluation of walk experience

The participants rated their experience during their walk and their enjoyment of the environment using the following four items: (1) I was fine during the walk, (2) It was a pleasant time, (3) I liked the route I went through, and (4) The sounds I listened to from my headphones bothered me. They were required to rate agreement or disagreement with these items using a 7-point Likert-type scale with anchors 1 = absolutely disagree and 7 = absolutely agree.

## Data analysis

The mean walking speeds were calculated for specific sections of the route. A two-way mixed analysis of variance (ANOVA) was conducted to access the effects of the condition (forest birdsong, crowded city noise, real outdoor noise), and the section of the route

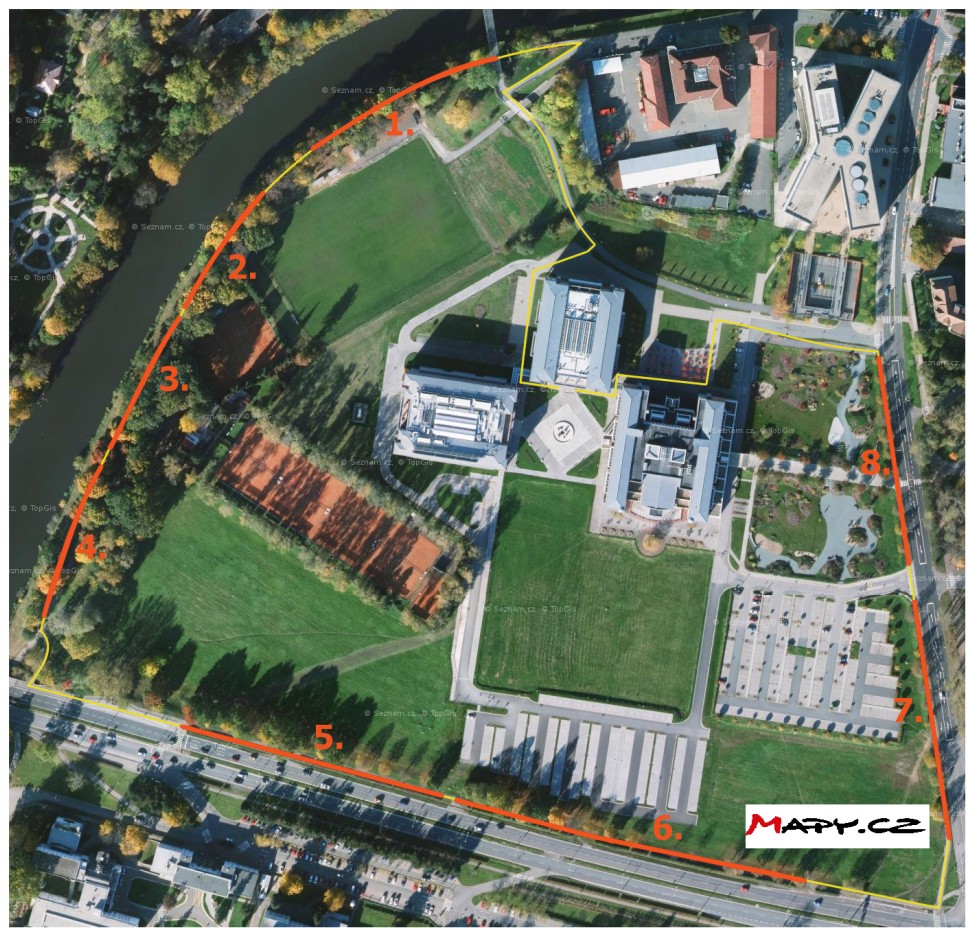

**Figure 1** **Sections 1–8, where walking speed was measured, are indicated.** Source: Mapy.cz, ©Seznam.cz, a.s.

(1–8) on walking speed. Greenhouse-Geisser correction was applied based on the test's Epsilon ($\varepsilon$) where assumption of sphericity was violated as assessed by Mauchly's test of sphericity. Differences between reported evaluations of the walk experience under particular conditions were compared by using a one-way ANOVA or $t$-test for independent samples. Some participants did not answer to questions related to the evaluations of the walk experience. They were excluded from the statistical analysis. Statistical analyses were conducted using IBM SPSS Statistics 24.

## Ethical statement

Ethical approval for the experiments was obtained from the Committee for Research Ethics at the University of Hradec Králové (No. 2/2018). All participants provided written informed consent. The participants signed a consent declaration in which they declared that they voluntarily participating in the experiment and that they were informed about the experimental procedure. They could decide to stop being a part of the research study

at any time without explanation. There were no known risks for participants in this study. The collected data were anonymised and used for the research purposes only.

## EXPERIMENT 1

### Methods

#### Participants

Eighty-seven undergraduates participated in the study. The students were young adults aged 19 to 23 years (M age = 20.51 yr., SD = 1.00), including 49 men and 38 women. They were recruited from a range of fields of study (informatics, financial management, and tourism) at the University of Hradec Králové. They were not paid to participate in the experiment.

#### Procedure

The participants were randomly assigned to a specific condition. The experiment was divided into one-hour blocks (Experiment 1–14 blocks, Experiment 2–16 blocks). Prior to the experiment, the participants registered for a particular block according to their availability. Gender, body height, and condition were balanced across each block to prevent the effects of immediate changes in atmospheric conditions or traffic density. Because not all preliminarily registered participants came to the experiment (due to illness, etc.), the sample sizes of the three conditions differed. Twenty-seven participants were assigned to the forest birdsong sounds condition, thirty-one participants were assigned to the crowded city noise condition, and twenty-nine participants were assigned to the real outdoor noise condition. The participants walked individually around the route. They were instructed to walk through the route at their normal walking speed. Furthermore, they were asked to not stop walking and not to call or speak with other people.

The walking route was marked by orange arrows painted on the sidewalk. The participants were asked to complete a questionnaire evaluating their experience after the walk. They were not informed about the goal of the study. Before the experiment began, the participants were asked whether they suffered from any current stress or anxiety (an upcoming exam, etc.). All of them gave a negative response.

The study was conducted in 2017 on three workdays: November 28 (in the afternoon), November 29, and November 30 (for details about atmospheric conditions see Table 1). The grass was green, bushes were yellow, and trees along the route were bare.

### Results

#### Analysis of walking speed

The mean time of walking for total route was 12.90 min (SD = 1. 28). The results revealed an overall faster walking speed under the crowded city noise condition (mean = 1.62 m/s, SD = 0.16) and the real outdoor noise condition (mean = 1.60 m/s, SD = 0.13) compared with the forest birdsong sounds condition (mean = 1.56 m/s, SD = 0.17). The mean walking speeds for specific sections of the route are shown in Table 2 and Fig. 2.

The mean walking speeds were calculated for specific sections of the route. A two-way mixed ANOVA was conducted to analyze the effects of acoustic conditions (forest birdsong,

**Table 1  Atmospheric conditions at the time of experiments.**

| Date | Part of day | Temperature | Cloudiness |
|---|---|---|---|
| | | Experiment 1 | |
| November 28 | Afternoon | 7 °C | Cloudy |
| November 29 | Morning | 2 °C | Cloudy |
| November 29 | Afternoon | 7 °C | Cloudy |
| November 30 | Morning | 4 °C | Cloudy |
| November 30 | Afternoon | 4 °C | Cloudy |
| | | Experiment 2 | |
| May 9 | Morning | 22 °C | Cloudy |
| May 9 | Afternoon | 23 °C | Cloudy |
| May 10 | Morning | 22 °C | Cloudy |
| May 10 | Afternoon | 24 °C | Cloudy |

**Table 2  The mean walking speeds (m/s) in specific sections of the route in Experiment 1.**

| Section | Forest birdsong | | Crowded city noise | | Real outdoor noise | |
|---|---|---|---|---|---|---|
| | Mean | SD | Mean | SD | Mean | SD |
| 1 | 1.53 | 0.18 | 1.61 | 0.20 | 1.59 | 0.14 |
| 2 | 1.54 | 0.20 | 1.60 | 0.16 | 1.58 | 0.15 |
| 3 | 1.55 | 0.19 | 1.59 | 0.19 | 1.59 | 0.13 |
| 4 | 1.56 | 0.20 | 1.61 | 0.18 | 1.60 | 0.14 |
| 5 | 1.58 | 0.17 | 1.63 | 0.16 | 1.61 | 0.15 |
| 6 | 1.59 | 0.18 | 1.65 | 0.22 | 1.61 | 0.13 |
| 7 | 1.58 | 0.17 | 1.62 | 0.17 | 1.60 | 0.12 |
| 8 | 1.59 | 0.20 | 1.64 | 0.17 | 1.61 | 0.13 |

crowded city noise, or real outdoor noise) and the route's environmental properties (the section of the route) on walking speeds. The acoustic condition was selected as the between-subjects factor, and the section of the route was selected as the within-subjects factor. The walking speed was selected as the dependent variable. The dependent variable was normally distributed for nearly all combination of the levels of the between-subjects and within-subjects factors (acoustic condition and section walked), as assessed by Shapiro–Wilk's test ($p > .05$), except for section 6 under the crowded city noise condition ($p = .007$).

A two-way mixed ANOVA indicated a statistically nonsignificant between-subjects main effect of the condition ($F_{2,84} = 0.886$, $p = .416$, $\eta2 = .021$). Furthermore, the ANOVA indicated a statistically significant within-subjects main effect of the section of the route ($F_{3.209,269.592} = 5.874$, $p < .001$, $\eta2 = .065$, $\varepsilon = .458$). A post hoc analysis with a Bonferroni adjustment showed that the participants' walking speeds differed statistically significantly in the following pairs of sections: 1–6, 1–8, 2–5, 2–6, 2–8, 3–5, 3–8, 4–8, and 7–8. There was no statistically significant interaction between the acoustic condition and the section of the route ($F_{6.419,269.592} = 0.604$, $p = .738$, $\eta2 = .014$, $\varepsilon = .458$).

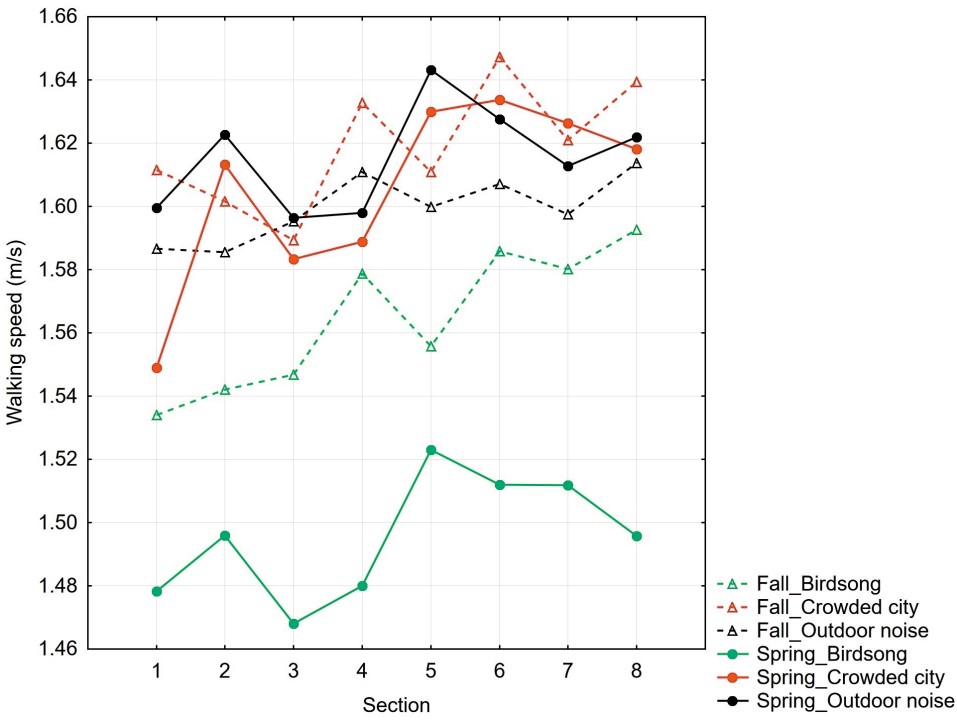

**Figure 2** **Mean walking speeds (m/s) for particular sections of the route in Experiment 1 and Experiment 2.** Dashed lines represent data from Experiment 1, conducted in November, and solid lines represent data from Experiment 2, conducted in May.

*Evaluation of walk experience*

The scores for particular items are listed in Table 3. It was examined how the agreement with the statement "I was fine during the walk" was related to the type of acoustic stimulus to which participants listened. One-way ANOVA did not indicate a statistically significant effect of the type of condition ($F_{2,70} = 2.30$, $p = .108$). One-way ANOVA indicated that agreement with the statement "It was a pleasant time" was statistically significantly influenced by the type of acoustic condition ($F_{2,70} = 5.211$, $p = .008$, $\eta2 = .13$). A post hoc Tukey test indicated significant differences between the forest birdsong and the crowded city noise conditions. The walk was a more pleasant experience for the participants who listened to forest birdsong sounds than for those under the crowded city noise condition. However, agreement with the statement "I liked the route I went through" was not influenced by the type of acoustic condition ($F_{2,70} = 1.042$, $p = .366$). A $t$-test for independent samples indicated significant differences between levels of agreement with the statement "The sounds I listened to from my headphones bothered me" under both conditions with soundtracks. The traffic noise bothered the participants more than the forest birdsong sounds ($t = 5.653$, $p < .001$, Cohen's $d = 1.581$).

## Discussion

The results showed some differences in average walking speeds under specific conditions, but these differences were small and statistically nonsignificant. The participants who

**Table 3 Evaluation of the walk experience in Experiment 1.** Measured by the level of agreement with particular items. The scale ranged from 1 to 7.

| Item | Forest birdsong | | Crowded city noise | | Real outdoor noise | |
|---|---|---|---|---|---|---|
| | Mean | SD | Mean | SD | Mean | SD |
| I was fine during the walk. | 6.04 | 1.11 | 5.50 | 1.44 | 6.17 | 0.78 |
| It was a pleasant time. | 5.77 | 0.95 | 4.54 | 1.44 | 5.26 | 1.60 |
| I liked the route I went through. | 4.80 | 1.32 | 5.33 | 1.24 | 5.17 | 1.40 |
| The sounds I listened to from my headphones bothered me | 1.65 | 1.16 | 4.62 | 2.39 | – | – |

listened to birdsong sounds walked slightly slower than the participants under the crowded city noise and real outdoor noise conditions, but in contrast to the previous experiment (*Franěk et al., 2018*), these differences were not significant. We assumed that the small effect of these acoustic stimuli was caused by incongruence between spring or early summer bird sounds and the appearance of the environment in late fall. The small and nonsignificant differences between the crowded city noise condition and the real outdoor noise condition likely reflect the fact that the outdoor environment was generally noisy; thus, listening to the crowded city noise soundtrack and hearing the ambient noise under the control condition did not result in a substantial difference.

The statistical analysis revealed statistically significant differences in walking speeds only among particular sections of the route that reflected both their environmental character and noise level. In general, there was a slower walking speed in sections 1–4, which were located in the oak alley, with a low level of noise. In contrast, participants walked faster in sections 5 and 6, which had a high level of noise and traffic. Finally, the participants statistically significantly increased their walking speed in the last section of the route. This increase may simply reflect the fact that they had already seen the end of the route and decided to quickly finish their task.

Although many differences in walking speed among the consequent sections are visible in the graphical representations of the data (Fig. 2), the post hoc test did not indicate statistical significance (except for the differences between sections 7 and 8). This suggests that these differences in walking speed may reflect tiny combinations of the effects of visual features of the environment, acoustic conditions, immediate changes in atmospheric conditions and individual variables. The difference between sections 7 and 8 could also be explained by the participants' intention to quickly finish their task.

The results also revealed that the type of acoustic stimulus to which participants listened during the walk had only a small effect on their evaluation of the walk experience because significant differences were found only for the statement "It was a pleasant time" between the forest birdsong and the crowded city noise conditions. These findings also reflect the effects of incongruence between the sounds and the environment.

**Table 4  The mean walking speeds (m/s) in specific sections of the route in Experiment 2.**

| Section | Forest birdsong | | Crowded city noise | | Real outdoor noise | |
|---------|------|------|------|------|------|------|
| | Mean | SD | Mean | SD | Mean | SD |
| 1 | 1.48 | 0.18 | 1.55 | 0.26 | 1.60 | 0.22 |
| 2 | 1.50 | 0.12 | 1.61 | 0.15 | 1.62 | 0.22 |
| 3 | 1.47 | 0.12 | 1.58 | 0.15 | 1.60 | 0.22 |
| 4 | 1.48 | 0.11 | 1.59 | 0.15 | 1.60 | 0.19 |
| 5 | 1.52 | 0.11 | 1.63 | 0.16 | 1.64 | 0.21 |
| 6 | 1.51 | 0.11 | 1.63 | 0.17 | 1.63 | 0.22 |
| 7 | 1.51 | 0.11 | 1.63 | 0.15 | 1.61 | 0.22 |
| 8 | 1.50 | 0.25 | 1.62 | 0.16 | 1.62 | 0.21 |

# EXPERIMENT 2

## Methods

### Participants

Sixty-five undergraduates participated in the study. The students were young adults aged 19 to 28 years (M age = 21.14 yr., SD = 1.00), including 29 men and 36 women. They were recruited from a range of fields of study (informatics, financial management, and tourism) at the University of Hradec Králové. They were not paid to participate in the experiment. Twenty three participants (11 females) took part also in Experiment 1. However, they were exposed to different acoustic conditions than they had in Experiment 1. Given that the walking route located close to the university should be known to all participants, we do not expect that the participation in Experiment 1 could affect the results.

### Procedure

The procedure was the same as the one used in Experiment 1. Because not all preliminary registered participants did come to the experiment (due to illness, etc.), the sample size of each of the tree conditions was different. Nineteen participants were assigned to the forest birdsong sounds, twenty-four participants were assigned to the crowded city noise condition, and twenty-two participants were assigned to the real outdoor noise condition.

The study was conducted in 2018 on two workdays: May 9 and May 10 (for details about atmospheric conditions see Table 1). The grass was bright green, and the trees had leaves

## Results

### Analysis of walking speed

The mean time of walking for total route was 13.02 min (SD = 1. 33). The results revealed an overall faster walking speed under the crowded city noise condition (mean = 1.61 m/s, SD = 0.15) and the real outdoor noise condition (mean = 1.62 m/s, SD = 0.20) compared with the forest birdsong sounds condition (mean = 1.49 m/s, SD = 0.11). The mean walking speeds for the specific sections of the route are shown in Table 4 and Fig. 2.

The mean walking speeds were calculated for specific sections of the route. A two-way mixed ANOVA was conducted to analyze the effects of acoustic conditions (forest birdsong, crowded city noise, or real outdoor noise) and the route's environmental properties (the

section of the route) on walking speeds. The acoustic condition was selected as the between-subjects factor, and the section of the route was selected as the within-subjects factor. The walking speed was selected as the dependent variable. The dependent variable was normally distributed for nearly all combination of the levels of the between-subjects and within-subjects factors (acoustic condition and section walked), as assessed by Shapiro–Wilk's test ($p > .05$), except for section 1 under the forest birdsong condition ($p = .002$).

A two-way mixed ANOVA indicated a statistically significant between-subjects main effect of the condition ($F_{2,62} = 3.290$, $p = .044$, $\eta2 = .096$). A Games-Howell post hoc analysis showed that the participants listening to the crowded city noise walked statistically significantly faster on the route than participants listening to forest birdsong sounds ($p = .026$). The participants in the real outdoor noise condition walked faster than participants listening to the forest birdsong sounds, but this difference was not statistically significant ($p = .067$). The participants in the real outdoor noise condition walked statistically non-significantly faster than participants listening to the crowded city noise ($p = .981$). The ANOVA indicated a statistically significant within-subjects main effect of the section ($F_{2.128, 131.915} = 4.818$, $p = .008$, $\eta2 = .072$, $\varepsilon = .304$). A post hoc analysis with a Bonferroni adjustment showed that the participants walking speeds differed statistically significantly in the following pairs of sections: 2–3, 2–4, 3–5, 3–6, 3–7, 4–5, 4–6, and 4–7. There was no statistically significant interaction between the acoustic condition and the section of the route ($F_{4.255, 131.915} = 0.476$, $p = .765$, $\eta2 = .015$, $\varepsilon = .304$).

### *Evaluation of walk experience*
The scores for particular items are listed in Table 5. It was examined how the agreement with the statement "I was fine during the walk" was related to the type of acoustic stimulus to which participants listened. One-way ANOVA indicated non-significant effect of the type of condition ($F_{2,58} = 2.727$, $p = .073$, $\eta2 = .08$). One-way ANOVA indicated that agreement with the statement "It was a pleasant time" was statistically significantly influenced by the type of acoustic condition ($F_{2,58} = 4.199$, $p = .020$, $\eta2 = .13$). A post hoc Tukey test indicated significant differences between the forest birdsong and the crowded city noise conditions. The walk was a more pleasant experience for the participants who listened to forest birdsong sounds than for the participants listening to the crowded city noise. However, agreement with the statement "I liked the route I went through" was not influenced by the type of acoustic condition ($F_{2,58} = 0.424$, $p = .66$). A $t$-test for independent samples indicated significant differences between the levels of agreement with the statement "The sounds I listened to from my headphones bothered me" under both conditions with acoustic stimuli. The crowded city noise bothered the participants more than the forest birdsong sounds ($t = 5.486$, $p < .001$, Cohen's $d = 1.858$).

## Discussion
In contrast to Experiment 1, the statistical analysis revealed a significant effect of the acoustic condition on walking speed. The participants who listened to birdsong sounds walked statistically significantly more slowly than the participants under the crowded city noise condition. Although we found that the mean walking speed was slower under the

**Table 5  Evaluation of the walk experience in Experiment 2.** Measured by the level of agreement with particular items. The scale ranged from 1 to 7.

| Item | Forest birdsong | | Crowded city noise | | Real outdoor noise | |
|---|---|---|---|---|---|---|
| | Mean | SD | Mean | SD | Mean | SD |
| I was fine during the walk. | 6.31 | 0.70 | 5.13 | 1.74 | 5.77 | 0.83 |
| It was a pleasant time. | 5.50 | 1.26 | 4.13 | 1.98 | 5.27 | 1.45 |
| I liked the route I went through. | 5.56 | 0.89 | 5.48 | 1.20 | 5.77 | 1.11 |
| The sounds I listened to from my headphones bothered me | 1.63 | 1.26 | 4.78 | 2.04 | – | – |

condition of birdsong sounds than under the real outdoor noise conditions, this difference was not statistically significant. The data plotted in Fig. 2 show clear differences between Experiment 1 and Experiment 2; specifically, there were substantially greater differences in walking speed under the birdsong sound condition and the other two conditions. Because this experiment was conducted in the spring, it is assumed that the birdsong sounds were perceived as more congruent with the spring environment and therefore had a greater relaxing effect than the same sounds presented in the fall, which resulted in a considerable decrease in walking speed on the route.

The analysis also showed the significant effects of the sections of the route. As expected, the walking speed was slower at the beginning of the route in the section located along an oak alley with a low level of traffic noise. The acceleration of the walking speed under all conditions in section 2 may be explained in terms of a possible negative reaction to a specific place in this section—a damaged fence painted with graffiti on the left side of the route. Although this location had the same appearance in both experiments, it may have been less visually congruent with the surrounding environment in beautiful spring vegetation. Consistently, in a study by *Franěk, Van Noorden & Režný (2014)*, in which the participants were asked to walk along a different route, they sped up close to a house with a damaged facade. As expected, in sections 5 and 6, which had the highest level of traffic noise, the walking speed increased, while in sections 7 and 8, which had lower levels of noise, the walking speed decreased. In contrast to fall, the participants did not speed up along the final section of the route. This difference might reflect the effect of the season; in springtime, because of the warmer temperatures and lively vegetation, the walk was more pleasant than that in fall; thus, the participants might not have intended to quickly finish their task.

## GENERAL DISCUSSION

This study examined the effects of listening to birdsongs or traffic noise on walking speed in a real outdoor environment in two different seasons, spring and fall. It was predicted that listening to the soundtrack with intense crowded city noise would increase walking speed, while listening to the soundtrack with natural bird sounds would decrease walking speed. Furthermore, it was predicted that the effect of bird sounds would be more salient in the spring than in the fall because of congruence between the sounds and the environment.

The results showed that listening to the soundtrack with birdsongs was associated with a slower walking speed than that under the soundtrack with crowded city noise or listening to real outdoor noise. However, the effect of the acoustic condition was statistically significant only in Experiment 2, which was conducted in spring, where we found a significant difference between the walking speed while listening to the soundtrack with birdsongs and the walking speed while listening to the soundtrack with crowded city noise. Moreover, the results did not show significant differences between the crowded city noise condition and the real outdoor condition. The results also revealed the effects of environmental features of particular areas of walking routes on walking speed. Regardless of the experimental condition, in areas with a relatively high natural character and low level of outdoor noise participants tended to walk slower than in areas with a lower number of natural elements and a higher level of noise.

Although there is a large body of studies documenting the various aspects of the positive effects of greenery and vegetation (for a review, see *Bowler et al., 2010*), there is a lack of studies examining the effects of ''greenery'' in late fall or in winter, when green elements are absent from the environment. A significantly greater effect of forest birdsong sounds on walking speed was found when compared to their effect on walking speed under the crowded city noise condition in the experiment conducted in May, but in the experiment conducted in November, the effect was small and statistically nonsignificant. This finding raises questions regarding the effect of the congruence between auditory and visual stimuli. It was hypothesized that the effect of nature sounds based on bird songs on walking speed would be more salient in the spring than in the fall because of the congruence between the sounds and the visual environment. Clearly, one may suppose that relaxation sounds based on spring or early summer forest birdsong are not congruent with fallen leaves, dark and cloudy skies and a low temperature. This finding further supports the idea that the effect of perceived congruence between sounds and the visual characteristics of an environment may play some role in an individual's reaction to the environment. This finding also raises the question of the effects of listening to different types of relaxation soundtracks or music while walking in different types of environments. Acoustic stimuli that are not congruent with the visual characteristics of an environment might not have a relaxing effect. Interestingly, similar findings showing congruence between season and another type of acoustic stimulus, preferred music, have also been found in recent research in music psychology. Experiments conducted by *Pettijohn, Williams & Carter (2010)* suggested that in fall and winter, people might prefer reflexive and complex music (e.g., classical music, jazz), while in spring and summer, they prefer rhythmic and upbeat and conventional music. Consistently, *Krause & North (2018)* reported listener preferences for arousing music in the warmer months, serene music in spring, and melancholy music in fall and winter. Moreover, research has also provided some evidence for seasonal variation in color preference (*Schloss & Heck, 2017*; *Schloss et al., 2017*). We can consider the practical applications of these outcomes. Recently, *Steele et al. (2019)* reported the effects of a ''Musikiosk Soundscape Intervention'' that allowed visitors of a small public park to play in a gazebo music from their own devices over publicly provided speakers. It was shown that the park was perceived as being more pleasant than it was prior to the intervention; moreover, the perceived soundscape

calmness and appropriateness were not affected. Some visitors also appreciated that music masks unpleasant traffic noise. It is worth considering using artificial natural sounds, such as birdsongs, in urban parks or along roads together with various forms of urban greenery or other natural elements to provide more intense restorative effects. However, the effect of possible incongruence between various stimuli should be taken into account.

While the mean walking speeds under the crowded city noise condition and the real noise condition were roughly identical in both experiments, under the forest birdsong sounds condition, the mean walking speed was faster in fall than in spring. Thus, a possible effect of seasonal temperature on walking speed is worth mentioning. One may suppose that people walk faster in colder temperatures to escape from cold outdoor environments. If so, our results could be influenced more by this effect than by the above discussed congruence between sounds and the environment. However, there is no empirical evidence for this assumption. In contrast, there is evidence that heat causes individuals to walk faster (e.g., *Rotton, Shats & Standers, 1990*) and that people have a greater normal walking speed in summer than in winter (e.g., *Montufar et al., 2007*). However, in both experiments, neither extreme heat nor extreme cold occurred.

In contrast with *Franěk et al. (2018)*, there were no substantial and significant differences between the walking speeds of the participants who listened to the soundtrack with crowded city noise from headphones and those of the participants in the real outdoor noise condition who did not listen any sounds from headphones. The explanation can be found in the difference in the acoustic conditions of the walking routes used in the two studies. The route used in the current study, which included sections located along a road with heavy traffic, was noisier than the route used in the previous experiment, specifically the sections located around the road with heavy traffic. Thus, participants assigned to the real outdoor noise condition listened to the real traffic noise from the outside environment. However, the traffic noise soundtrack "Hectic Kolkata" represented city traffic that was much more hectic and stressful than the real acoustic environment of the walking route; moreover, it contained both traffic noise and speech, but it did not result in a faster speed compared to that in the real outdoor noise condition. We might suppose that this soundtrack might increase arousal more than listening to noise in the real outdoor environment, but the results did not show differences in walking speed between the two conditions. These findings raise questions regarding the effects of diverse types of noises on walking speed. Obviously, there is no linear-like association between the level of traffic noise and its arousing effect and pedestrian walking speed. Traffic noise may increase walking speed to a certain level, but it seems that the upper comfortable pace of walking speed should limit the pace of behavioral response to the noise (*Bohannon, 1997*). Only very intense and unpleasant sounds could cause a panic-like reaction that prompted the participants to move as quickly as possible to get out of the area.

It is worth speculating why people might walk faster in the presence of traffic sounds. We already mentioned the explanation assuming that a fast pace of life, including walking speed, may be a response to stimulatory overload and various urban stressors, including crowding and traffic noise (e.g., *Bornstein & Bornstein, 1976*). Investigations of walking speed have been conducted in down-town areas, and such studies (e.g., *Bornstein & Bornstein, 1976*;

*Kirkcaldy, Furnham & Levine, 2001*; *Levine & Bartlett, 1984*; *Lowin et al., 1971*; *Walmsley & Lewis, 1989*) documented that people walk faster in large cities than in small towns. However, these studies did not precisely analyze the effects of specific acoustic and visual properties. Although we know that noise is an environmental stressor (e.g., *Evans, 1984*) and its arousing effects might activate the internal clock, which may result in the acceleration of walking speed, we also offer an alternative explanation in terms of an older theory that has been frequently applied in the field of environmental psychology, specifically the Mehrabian-Russell theory of approach-avoidance behavior (*Mehrabian & Russell, 1974*). Here, the authors described two forms of affective response to an environment. Approach behavior means that the individual is trying to establish contact with the environment and stay inside it, while avoidance behavior means that he/she is trying to avoid such contact and to move away. According to the theory, the consequent behavioral reaction consists of various forms of behavior, such as physical movement heading into the environment or out of it, environment exploration, attention given to the environment, etc. This theory has been tested, namely, in consumer research (for a review, see *Bitner, 1992*). In addition, our previous study (*Franěk, 2013*) documented that accelerations in walking speed in sections of a route with a relatively low amount of greenery and high level of noise were linked with avoidance behavior (measured by dimensions of pleasure, arousal, and dominance from the Mehrabian-Russell questionnaire). Thus, if noise is not perceived as an actual stressor, it may make an environment less pleasant and less attractive, which results in the intention to move away and accelerate walking speed. On the other hand, we know from consumer research that merchants use various stimuli, including acoustic stimuli (music), to attract consumers to the marketplace and influence them to remain there for a long time (for a review, see *Jain & Bagdare, 2011*).

Furthermore, the interactions among the effects of the specific environment of the route, walking speed and the presented sounds were analyzed. In previous studies of walking speed (*Franěk, 2013*; *Franěk, Van Noorden & Režný, 2014*), it was observed that people walked relatively slowly in areas with more natural elements, namely, in dense alleys. The first four sections of the current walking route were of a highly natural character. Under almost all conditions in both experiments, slower walking speeds were observed in these sections. However, the natural aesthetic of section 2 was disturbed by an unaesthetic element—a damaged fence painted with graffiti on the left side of the route—which led to an increase in walking speed in Experiment 2. It should be noted that the results were more robust for the interactions between the effects of the specific environment of the route and walking speed than for the interaction between the presented sounds and walking speed. These findings support previous findings that the positive effect of an environment with a high number of natural features may result in a decrease in walking speed.

As predicted, the types of sounds experienced also influenced the ratings of visual environments. However, in our previous study (*Franěk et al., 2018*), we found that listening to forest birdsong improved various facets of the participants' walk experience (a good feeling during the walk, liking the route, the walk was a pleasant time) in comparison to the other acoustic conditions, while in the current experiments in both fall and spring condition we only found that the walk was a more pleasant time for participants who

listened to forest birdsong than crowded city noise. It should be noted that the current walking route was generally less pleasant than the route used in the previous experiment. Moreover, while the former route was located in relatively calm green areas (a small park; a meadow; a long, dense oak alley), and the experiment was carried out in spring, the current route encompassed natural areas only at its beginning. All these factors further speak to the effect of audio-visual congruence. Clearly, forest birdsongs are not very congruent with the sight of an urban road with heavy traffic.

The present study has some limitations. The first limitation is that the experiment did not include a true control condition. Walking without hearing any soundtrack was not a true control condition, because participants were still exposed to some outdoor noise. However, data from this investigation can be compared with the findings obtained in the previous study (*Franěk et al., 2018*) conducted in a relatively calm area with the same research methodology.

The majority of research on the restorative potential of environments has been conducted in the laboratory. Conducting such research in an outdoor environment the observation of behavior in real conditions, but the drawback of the methodology used is that it was not possible to control for all external variables, such as immediate changes in traffic density, weather and atmospheric conditions. However, differences between laboratory experiments and studies conducted in real outdoor environments in research on the restorative effect of the natural environment are worth evaluating. Although conclusions from an early meta-analysis conducted by *Stamps (1990)* suggest that photographs have representational validity in regard to the real outdoor world, some more recent studies might question this assumption (e.g., *Daniel & Meitner, 2001*). In their methodological consideration, *Conniff & Craig (2016)* propose that the most ecologically valid situation may be to conduct research in situ in real outdoor conditions. Thus, the ecological validity of this study is its strength, and the drawback is the lack of full control of all external variables. Therefore, we tried to reduce the effects of these variables by balancing the experimental conditions during the days on which the experiment was performed.

It is also worth mentioning that we had different sample sizes and different numbers of male and female participants in the two experiments. It is possible that the smaller sample size in Experiment 2 resulted in smaller effect sizes and a nonsignificant difference between the mean walking speed under the real outdoor noise condition and the forest birdsong sounds condition. However, regarding the different numbers of male and female participants in the two experiments, there is no evidence concerning the effect of gender on noise sensitivity (e.g., *Ellermeier, Eigenstetter & Zimmer, 2001*).

Furthermore, it was not considered individual differences in noise annoyance or attitudes toward particular types of noise. Although we did not test the personality traits of the participants in this sample, our previous investigations conducted with students in the same fields of the study showed that they were slightly less neurotic and slightly more extraverted compared to the age-matched general population in the Czech Republic (*Franěk, Van Noorden & Režný, 2014*). Whereas lower neuroticism and higher extraversion are generally associated with lower noise annoyance (*Dornic & Ekehammar, 1990*), it can be predicted that the negative reactions of our sample on noise were smaller than those

of a general population. However, the scores on these personality dimensions could differ among individual participants, which could be a source of uncontrolled variability. Moreover, recent research has shown that not only personality traits but also attitudes toward particular sources of noise can influence evaluation of the source of noise and its appropriateness in a specific environment. For instance, *Taff et al. (2014)* found that educational messaging affected visitors of national parks to accept military aircraft sounds. Recently, *Benfield et al. (2018)* showed that attitudes in favor of motorized recreation and attitudes in favor of the regulation of motorized recreational noise can alter the effects of motorized recreation noise on scenic evaluations in opposing directions. Thus, we can assume that attitudes in favor of motorized recreation or in favor of pristine nature could affect the evaluation of traffic noise and consequent behavioral reactions.

A further factor that could influence the observed behavior might also be the cultural background of the participants. Clearly, living and experience in different physical and acoustic environments may affect reactions to the specific environment where these experiments were conducted. To prevent this possible effect, we selected only participants with similar cultural backgrounds. They were all of Czech nationality and originated from neighboring regions.

Finally, the study was conducted with a young-aged sample. Clearly, young people walk faster than older ones (*Bohannon, 1997*); thus, the walking speeds observed in this study are typical of young healthy people. Different results may be found for older people or for people with some physical handicaps

## CONCLUSIONS

In conclusion, the study explored the interaction between auditory and visual stimuli in the perception of an environment and showed consequent behavioral reactions, specifically changes in walking speed. Although noise may increase walking speed, the present experiments also suggest that listening to nature sounds (forest birdsongs) from headphones while walking along an urban route may decelerate the walking speed. In accordance with previous studies, our study also showed that perceived congruence between the acoustic and visual stimuli also plays a role. Spring and early summer bird songs that are not congruent with the season and vegetation period have smaller and nonsignificant effects than congruent acoustic stimuli. Moreover, it was also shown that exposure to noise may influence the perception of an environment. The same environment may be more liked in the absence of noise or in the presence of relaxation sounds. The results raise questions about the restorative function of urban greenery in different seasons. Our findings also provide some practical implications. For instance, artificial natural sounds, such as birdsongs in urban parks or along roads can be used to provide together with various forms of urban greenery or other natural elements more intense restorative effects.

## ACKNOWLEDGEMENTS

We thank Nicol Klapalová, Vít Brouček, Jan Freiberg, František Linder, and Daniel Hejduk for their help in organizing and conducting the experiments.

### Funding

This work was supported by the Student Specific Research Grant 1/2018 from the Faculty of Informatics and Management at the University of Hradec Králové. The funders had no role in study design, data collection and analysis, decision to publish, or preparation of the manuscript.

### Grant Disclosures

The following grant information was disclosed by the authors:
Student Specific Research Grant: 1/2018.
Faculty of Informatics and Management at the University of Hradec Králové.

### Competing Interests

The authors declare there are no competing interests.

### Author Contributions

- Marek Franěk conceived and designed the experiments, performed the experiments, analyzed the data, contributed analysis tools, prepared figures and tables, authored or reviewed drafts of the paper, approved the final draft.
- Lukáš Režný conceived and designed the experiments, performed the experiments, analyzed the data, contributed analysis tools, authored or reviewed drafts of the paper, approved the final draft.
- Denis Šefara analyzed the data, approved the final draft.
- Jiří Cabal analyzed the data, contributed analysis tools, approved the final draft.

### Human Ethics

The following information was supplied relating to ethical approvals:

Ethical approval for the experiments was obtained from the Committee for Research Ethics at the University of Hradec Králové (No. 2/2018).

### Data Availability

The raw measurements are available in the Supplemental File.

The acoustical stimuli are also available in figshare: Rezny, Lukas; Franěk, Marek; Šefara, Denis; Cabal, Jiří (2019): Birdsongs. figshare. Media. https://doi.org/10.6084/m9.figshare.9699230.v1

Rezny, Lukas; Franěk, Marek; Šefara, Denis; Cabal, Jiří (2019): Crowded city traffic noise. figshare. Media. https://doi.org/10.6084/m9.figshare.9105128.v1.

## Supplemental Information

Supplemental information for this article can be found online at http://dx.doi.org/10.7717/peerj.7711#supplemental-information.

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
