# Peer review of "Effect of birdsongs and traffic noise on pedestrian walking speed during different seasons"

_PeerJ, doi:10.7717/peerj.7711_

## Round 0.1 · original submission · Major Revisions

The three reviewers all agree that the draft needs substantial (as well as less radical) interventions. I believe that your manuscript should be carefully revised taking into account all of the requests, suggestions and comments, and assume that all in all they might appear overwhelming but at the end of the day they should in fact be manageable. I also believe that a second round of external review will be in any case necessary in order to assess whether the manuscript will have objectively improved following your revision (and a detailed description of the changes ad amendments it will bring) at all leves, from the many language issues to the more theoretical and methodological aspects raised, particularly by Reviewer 3. I am thus not guaranteeing that a positive decision will follow automatically from your revision, but invite you to consider carefully all of the many constructive and useful points raised by the reviewers (notice that Reviewer 1 also included an annotated text which is not identical to the review).

Reviewer 1 ·

Basic reporting

The article uses clear and technically correct text. On the introduction, some paragraphs could be rearranged to improve the text flow. See my comments on the uploaded file.

The article includes sufficient background to demonstrate how the work fits into the broader field of knowledge. Nevertheless, some results of previous studies were presented as showing the expected effect even if they were not statistically significant. This could lead to misunderstandings.

The structure of the article is conforming to an acceptable format. Figures are relevant to the content of the article, but Figure 1 do not has a reference from where it was taken from.

Please see my comments about tables’ labels on the upload file.

The raw data have been made available. I suggest to provide information about time of walking for each section (1– 8) and for total route in the raw data.

The authors claimed that the work is self-contained justifying it in an appropriate manner.

Experimental design

The research question is well defined, relevant and meaningful. The authors intend to fill the knowledge gap regarding the evaluation of direct behavioral responses (walking speed) to noise in specific environments.

The authors affirm that there were no known risks for participants in this study. Nevertheless, any potential risks of walking around the route and the discomfort it may entail should be considered for ethical purposes.

The soundtrack link of the relaxation sound condition was not correctly typed. The stimuli were not described with sufficient detail to replicate and there is no credit attribution for the authors of the soundtracks.

The authors have presented the effect size value for the analysis. The sample size of each of the three conditions was different and it was not specified in the text for the analysis.

Validity of the findings

The authors did not state that the data were made available in an acceptable discipline-specific repository but they uploaded them as Supplemental files.

The data was not fully controlled because the research was conducted in an outdoor environment. Nonetheless, according to the authors they have tried to reduce the effects of the external variables by balancing the experimental conditions during the days on which the experiment was performed.

The results were more robust for the interactions between the effects of the specific environment of the route and walking speed than for the interaction between the presented sounds and walking speed.

In some parts of the manuscript, the authors attribute that there was a difference between two conditions even if this difference was not statistically significant. This should be considered very carefully.

The conclusions are appropriately stated but in some cases are not supported by the results, or are overestimated. Some of the hypotheses presented in the introduction were not taken up during the discussion of the results.

Additional comments

Please see my comments in the uploaded file.

Annotated reviews are not available for download in order to protect the identity of reviewers who chose to remain anonymous.

Reviewer 2 ·

Basic reporting

Reporting is good throughout. some additional references are suggested below.

Experimental design

Research question well defined, relevant & meaningful. It is stated how the research fills an identified knowledge gap.
Methods described with sufficient detail & information to replicate.

Validity of the findings

Impact and novelty could be assessed more deeply (see suggestions below on implications and relevance).

Additional comments

The paper explores the effects of relaxation nature sounds (birdsong) compared to traffic noise on walking speed in a real outdoor urban environment. This is an interesting and novel topic that has important implications for policy and practice. The methodology is robust and the reporting is very precise and high quality. I suggest below some changes that would help to make the article stronger.
First of all, while the title refers to relaxation nature sounds, I recommend to change this into birdsongs and make explicit reference from the title and abstract that the sound considered was that of birdsongs. This can help readers to identify the topic of the study more clearly.

Abstract

Suggest to remove “Although there is a large body of studies based on subjective evaluations of noise annoyance or environmental preference” and focus on what the study is about.
Since one of the main points of the study was to test congruence, I suggest to make this clearer in the abstract earlier on.
Please add: The participants (n=x), young university students, walked along a 1.8-km urban route in the city of xx.
Please remove because the outdoor environment was, in general, quite noisy. – the word because is misleading as it is not possible to assess causality. Same suggestion for: because of the congruence between the sounds and the visual environment. – please rephrase avoiding word because.

Introduction

The section gives context, but the start doesn’t do a great job in identifying what is the core topic of the article. Since the article is on sounds, I suggest starting it with background information on the role of sound in restorative experiences, especially on birdsongs.

Also, since the study is about stress and walking in cities, I suggest to add background information on the relevance of these topics. For example, some references to urbanisation trends, or to the need to encourage walking and improving psychological wellbeing of city residents. These additions would reinforce the relevance and significance of the study.

Line 179
Regarding the negative health consequences of the fast pace of walking, I suggest to add further references, including some more recent research. This would reinforce the claim that high walking speed is detrimental for health.

Methods

Please add info on the city where the studies were performed.
Were Incentives/rewards offered to students?

I suggest to add one table with dates and weather conditions of each day. This would make reading easier.

Discussion

The start of the section needs more detail. I suggest that you improve the description at lines 543-547 to provide more detail on the main findings.

Regarding the congruence (lines 569 and following), I suggest to deepen the discussion on the practical implications of this matter. For example, one practical implication could be related to the use of relaxing artificial natural sounds in urban roads such as birdsongs: the findings seem to suggest that such artificial sounds should match weather and seasonal conditions to aid relaxation.

Line 574: it would be useful to re-state the finding on speed and season.

Lines 597-601 are unclear, I suggest to rephrase and reorganise.

Line 614: I suggest to change to The types of sounds experienced might have also influenced the ratings of visual environments. This is because it is hard to assess causality between the two, also due to external factors that were not controlled for.

It would be worth including some reflections and related literature on why people might walk faster in presence of traffic sound. This could be due to environmental stressors (Evans 2003) and safety concerns for example (Bornioli et al., 2019).

Limitations:
it seems that the order of walks was not randomised. If this is the case, I suggest to discuss the limitations and implications of this choice.
I also suggest to expand lines 632-636 and reflect more deeply on the strengths of a real-world environment. Is there any study that has compared relaxation effects between field and lab environments? See for example Conniff and Craig, 2016; Shaw et al., 2015 on relaxation and sensorial experience.

I suggest to add recommendations for future research, for example on:
- Research on benefits of nature and the role of seasons;
- Congruence between visual and audio stimuli and effects on behaviours and wellbeing;
- Personal characteristics and effects on walking speed and relaxation.

Conclusions are well stated, linked to original research question & limited to supporting results. I suggest to add recommendations for policy (for example see above on the use of artificial birdsong sounds in cities).


Minor changes:

Throughout text: When using the word significantly, I suggest to specify statistical significance.
Line 45 and 179
The research? Is it previous research or the current research?
Lines 45-57
Please add references to support your statements.
Line 544
There is something missing: First… (there is no “second”?)
Line 559
Suspicious? Not sure that’s the right word?
Table 2: title seems not correct as the table reports rating on walk not on speed?
For tables on ratings on walking experience, I suggest to add histograms that would also show statistical differences between ratings.

·

Basic reporting

The study reports the results of an experiment involving the effect of visual and auditory stimuli to relaxation in walking behaviour. The author looks at relaxation by means of direct simple individual reports (questionnaires) and walking speed across different urban environment settings. The study acknowledges that it is part of a sequence of studies related to walking, sound and urban environment.

As a study that is part of a larger initiative, the conceptual proposal of the experiments seems coherent and previous findings support part of the assumptions used by the authors. The experimental setup is not complex but let many possible interfering variables take part actual results. Most of the hypotheses and questions seem to reproduce other studies and the majority of the results reported are not conclusive, regarding the initial hypotheses. My general opinion is that the text could only be accepted if the authors promote large changes in the content, review of the literature and report of results.

Thought not-significant results should be reported and should be taken as an important part of the scientific development, the text and statistical reporting fail in providing a creative exploratory view of the data and information. Distributions, variability and possible cross-correlations between tendencies in the data are not clearly reported. Tendencies are hidden and the authors seem to avoid discussing null-hypotheses. This bleak report of non-significant data just reinforces some experimental design problems and hide insights that could help the reader.

I find problematic the definition (or lack of definition ) of the concept of relaxation behind the text. Although relaxation and stress have been studied for a long time, the authors do not discuss it from the literature, alternative definitions, or cultural variability. The choice for the stimuli seems to represent a culturally specific notion of relaxing sound. There is no problem in choosing an urban, Western, 2020's concept of relaxing sound. The problem here is that the study does not acknowledges the cultural, social, historical and psychological limitations of it, while makes culturally-centric choices. Since the concept of relax is very important for the study, my opinion is that it is not possible to understand the work without a clear review of the definition.

Following the problems of definition of relaxation and stress, the stimuli used as traffic noise seem to be very problematic. My guess is that the track includes an abnormal density of traffic noise. In other words, I don't think the surroundings of the experimental location exhibit, at anytime or context, the level of noise exhibited in the stimuli. It might impose a kind of cultural mismatch since, most probably, the subjects never experienced a similar context. It was recorded in a totally different cultural and urban environment and also includes an unreported/not-discussed concept of noise. It includes voices from different languages which, create another layer of psychological aspects of "noise" (in this case, traffic noise and speech).

Experimental design

The overall experimental design is feasible but the control of variables and consequent definition of limitations are weak. It does not mean that researchers must control all the aspects of the experiment (real experiments) but it seems that the authors and the statistical procedures ignored limitations coming from cultural assumptions, variability and uncontrolled variables that have an effect on the results. If the authors want to use ANOVA as a test for significant differences I strongly recommend discussing if the Anova's assumptions are violated. Another option is just use anova or other descriptive methods, as data visualization. Given the current state of the variability of the data that are reported and not reported, no test of significance can be easily applied. First, because there is no need for it (significance is useless in such a heavily influenced cultural behaviour), and second, because it is almost impossible to accept the assumptions of these methods.

Regarding the selection of participants, please, report if the participated in the previous studies and some better descriptions of their cultural background.

line 203: Since the previous study raises important questions tackled by the present one, the authors should be more specific and less simplistic concerning the overall design and results. For example:
- 203: " the sound variable was fully controlled": it is very difficult o control sound in the open environment. Please, specify the method
- 2017: "The participants listening to traffic noise walked
208 significantly faster on the route than those listening to forest birdsong sounds". Please, report what means significantly fast.
218: " the walking route was located in green areas with a relatively
219 small amount of traffic." I find "green areas and small amount of traffic" a very subjective definition that should be specified

237: "On the other hand, it is almost impossible to have a perfect control condition in a real outdoor environment.". I would also indicate that it is almost impossible to have similar treatment conditions in a real outdoor environment. How this impact in the assumptions required in the statistical methods? (e.g. how can you assume that deviations come from random processes?)

Validity of the findings

See discussion above in item 2

638: "Furthermore, it was not considered individual differences in noise annoyance or attitudes toward particular types of noise." I understand that the authors envisaged this problem. But it deserves a clear discussion using proper literature as base.

657 "In conclusion, the study demonstrated the interaction between auditory and visual stimuli in the perception of an environment and showed consequent behavioral reactions, specifically changes in walking speed." I think this conclusion is not supported by evidence. Authors should avoid typical terminology used in hard sciences and acknowledge uncertainty. I think the data explores, illustrates or suggests some relationships but the results (from a traditional statistical perspective) are not conclusive.

Additional comments

The content first paragraph is somewhat loose and seems to be repeated in the following paragraphs.

Lines 134, 137: "e.g." abbreviations should be placed inside the parentheses.

179: I cannot fully understand what means "pace of life" since life is very difficult to define or measure. Apart from the vagueness of the expression, it seems that there are cultural biases that seem to present an alternative slow pace of life as a generalizable solution. This may only reflect the viewpoints of subjects living in urban regions. Individuals from rural areas might have a different opinion or different problems with walking pace, the speed of changes, events and responses from the environments. This bias was also reproduced in the selection of subjects, who were probably selected from a group living in an urbanized area. Please, clarify or acknowledge the viewpoint.
244: Typo in "with to "..

---

## Round 0.2 · Minor Revisions

Clearly, all of the potential reorganization of text and last check for typos that have been indicated as necessary should be undertaken, although I would also suggest of paying great attention to the current comments and suggestions that the two surviving reviewers have formulated on less superficial issues (i.e. removing statistical overstatements, better specifying your position on fast walking and health). I believe that the sum of all this amounts to a minor revision, so it should be rather easily doable and in your immediate reach, and I look forward to seeing a revised version of your manuscript.

Reviewer 1 ·

Basic reporting

The article uses clear and technically correct text. The introduction was reorganized and it is clearer now. My only suggestion is to join the sections “Auditory and visual interactions: Sounds can change perception of environment”, “Auditory and visual interactions: Environment can change evaluation of sounds” and “Attempts to explain the phenomenon of auditory and visual interactions” in a single section, and include the section “the aims” in the “The current study” section.

The article includes sufficient background to demonstrate how the work fits into the broader field of knowledge, and the results from previous studies were adequately presented.

The structure of the article is conforming to an acceptable format. Figures are relevant to the content of the article, and Figure 1 has now been referenced.

The time of walking for each section and for total route were now presented in the raw data.

The authors claimed that the work is self-contained justifying it in an appropriate manner.

Experimental design

The research question is well defined, relevant and meaningful. The authors intend to fill the knowledge gap regarding the evaluation of direct behavioral responses (walking speed) to noise in specific environments.

The authors had affirmed that there were no known risks for participants in this study. Nevertheless, any potential risks of walking around the route and the discomfort it may entail should be considered for ethical purposes. The authors stated that the place is safe and located around the university campus. They consider that the participants were young healthy adults, who use to walk, and that the short walk on a flat surface did not cause any discomfort.

I still suggest uploading the stimuli to an online repository, to be consistent with the PeerJ’s policies.

The link for the “Forest Birdsong - Relaxing Nature Sounds - Birds Chirping” -https://www.youtube.com/ watch?v=Qm846KdZN_c - is not working because there is a missing letter. Please try this one: https://www.youtube.com/watch?v=Qm846KdZN_c
The authors have presented the effect size value for the analysis. The sample size of each of the three conditions was now specified in the text for the analysis.

Validity of the findings

The authors did not state that the data were made available in an acceptable discipline-specific repository but they uploaded them as Supplemental files.

The data was not fully controlled because the research was conducted in an outdoor environment. Nonetheless, according to the authors they have tried to reduce the effects of the external variables by balancing the experimental conditions during the days on which the experiment was performed.

The results were more robust for the interactions between the effects of the specific environment of the route and walking speed than for the interaction between the presented sounds and walking speed.

In some parts of the manuscript, the authors had attributed that there was a difference between two conditions even if this difference was not statistically significant and had overestimated some results. The authors have now corrected it and considered these statements more carefully. Nevertheless, they still present statistically non-significant results as a tendency to a certain result, not considering that the results are just either statistically significant or not. In other words, either the participants walked faster or not, because there were no statistically difference between groups. Instead, it could be discussed regarding the effect size as they have done for some issues in the discussion section.

The authors have now considered all of the presented hypotheses in their discussion.

Additional comments

Tha manuscript was muc improved. Please see my comments above for a few suggestions.

Reviewer 2 ·

Basic reporting

Please proofread the document, as there are some typos (since there are no page numbers or line numbers in the tracked changes version, it is hard to give exact locations of these typos).

I still recommend to for tables on ratings on walking experience to add histograms that show statistical differences between ratings.

Experimental design

no comment

Validity of the findings

I am still not convinced about the claim that walking fast has negative health consequences. I suggest to make more explicit the hypothesised implications of walking faster. Is walking faster bad for health? The authors mentioned some old research that walking faster might have health risks, but this is not enough in my opinion to say walking fast = negative health consequences. It would be ideal to give a figure of what is the ideal average walking speed (from previous literature).
Or, do results imply that since people walked faster in traffic environments, they are less likely to go out and walk in nowadays cities? This is a bit of a long shot. Also, moderate-intensity physical activity has positive health impacts. I suggest to add reflections on the meaning of "walking fast" in the limitations section.

Additional comments

I thank the authors for their hard work in editing the manuscript.

---

## Round 0.3 · accepted · Accept

I have appreciated the efforts made in oder to reply to the Reviewers' requests and suggestions, including the language revision, which have made overall the manuscript suitable for publication.

#